# Ethanolic Extract from *Limonia acidissima* L. Fruit Attenuates Serum Uric Acid Level via *URAT1* in Potassium Oxonate-Induced Hyperuricemic Rats

**DOI:** 10.3390/ph16030419

**Published:** 2023-03-09

**Authors:** Rika Yusnaini, Rosnani Nasution, Nurdin Saidi, Teti Arabia, Rinaldi Idroes, Ikhsan Ikhsan, Rahmad Bahtiar, Muhammad Iqhrammullah

**Affiliations:** 1Graduate School of Mathematics and Applied Sciences, Universitas Syiah Kuala, Banda Aceh 23111, Indonesia; 2Department of Psychology and Nursing, Faculty of Medicine, Malikussaleh University, Lhokseumawe 24351, Indonesia; 3Department of Chemistry, Faculty of Mathematics and Natural Sciences, Universitas Syiah Kuala, Banda Aceh 23111, Indonesia; 4Department of Agricultural, Universitas Syiah Kuala, Banda Aceh 23111, Indonesia; 5Department of Pharmacy, Faculty of Mathematics and Natural Sciences, Universitas Syiah Kuala, Banda Aceh 23111, Indonesia; 6Department of Surgery, Tgk. Chik Di Tiro Hospital, Sigli 24116, Indonesia; 7Pharmacology Laboratory, Faculty of Veterinary Medicine, Universitas Syiah Kuala, Banda Aceh 23111, Indonesia; 8Innovative Sustainability Lab, PT. Biham Riset dan Edukasi, Banda Aceh 23243, Indonesia

**Keywords:** creatinine, natural product, in vivo, phytocompound, uric acid

## Abstract

A high prevalence of hyperuricemia among adult and older adult populations has intrigued the development of its therapy based on natural products. Our objective was to investigate the antihyperuricemic activity of the natural product from *Limonia acidissima* L. in vivo. The extract was obtained through the maceration of *L. acidissima* fruits using an ethanolic solvent and was tested for its antihyperuricemic activity against potassium oxonate-induced hyperuricemic rats. Serum uric acid, creatinine, aspartate aminotransferase (AST), alanine aminotransferase (ALT), and blood urea nitrogen (BUN) were observed before and after the treatment. Expression of urate transporter 1 (URAT1) was also measured using a quantitative polymerase chain reaction. Antioxidant activity based on a 2,2-diphenyl-1-picrylhydrazyl (DPPH) scavenging assay, along with total phenolic content (TPC) and total flavonoid content (TFC), were measured. Herein, we present the evidence of the serum uric acid lowering effect of the *L. acidissima* fruit extract along with improved AST and ALT (*p* < 0.01). The reduction of serum uric acid was in accordance with the decreasing trend of URAT1 (1.02 ± 0.05-fold change in the 200 mg group), except in a group treated with 400 mg/kg body weight extract. At the same time, BUN increased significantly in the 400 mg group (from 17.60 ± 3.286 to 22.80 ± 3.564 mg/dL, *p* = 0.007), suggesting the renal toxicity of the concentration. The IC_50_ for DPPH inhibition was 0.14 ± 0.02 mg/L with TPC and TFC of 143.9 ± 5.24 mg GAE/g extract and 390.2 ± 3.66 mg QE/g extract, respectively. Further studies should be carried out to prove this correlation along with the safe concentration range of the extract.

## 1. Introduction

A recent meta-analysis (2009–2019) published in 2020 revealed that 14.4% adults suffer from hyperuricemia worldwide [1]. As a polygenic disease, hyperuricemia may develop into hypertension, kidney failure, kidney stones, gouty arthritis, and even cardiovascular disease as the consequence of uric acid crystals formation [2,3]. Most of the filtered uric acid in the glomerulus of the kidney will be reabsorbed (90%) in the tubules, mediated by specific transporter molecules such as urate transporter 1 (URAT1) [4]. This molecule functions as the major mechanism for regulating blood urate levels [4]. As the current first-line therapy, xanthine oxidase inhibitors, such as allopurinol, topiroxostat, and febuxostat, are used to lower uric acid levels [5,6]. Nonetheless, these uric acid lowering agents have adverse effects, including drug hypersensitivity [7] and looseness, hepatitis, and interstitial nephritis [8]. In this light, researchers have turned to natural products as an alternative to manage hyperuricemia. In rats with hyperuricemia, serum uric acid was reduced significantly by 38.76% after oral administration with 5 mL/L of orange juice [9]. The authors associated the anti-hyperuricemic effect of orange juice with the presence of hesperidin and narirutin [9]. Hydrolyzed extracts from *Adenanthera avonine* L. leaf, *Blumea balsamifera* (L.) DC. Leaf, *Morinda citrifolia* L. root, *Orthosiphon aristatus* (Blume) Miq. Leaf, *Peperomia pellucida* (L.) Kunth leaf, *Syzygium aromaticum* (L.) Merr., and L.M.Perry flower bud were found active as xanthine oxidase inhibitors (EC_50_ = 264.25 ± 0.64 to 39.58 ± 0.10 μg/mL) [10]. Statistical significance of plasma uric acid reduction was found in rats treated with ethanolic extract from *Marantodes pumilum* leaf in a time-dependent manner [11].

Oxidative stress imbalance was reported to cause renal abnormalities associated with the increased level of serum uric acid [12]. Xanthine oxidase produces uric acid as the end-product concomitant to oxidative hydroxylation of hypoxanthine to xanthine [13]. Thus, xanthine oxidase inhibition therapy in combination with antioxidant properties is considered strategic to manage hyperuricemia [5,13]. *Limonia acidissima* is one of the underutilized medicinal plants with antioxidant potential, which has been reported multiple times [14,15]. Antioxidant potential based on various in vitro screening assays of hydroalcoholic extract from *L. acidissima* L. fruit have been reported the highest as compared with those obtained through chloroform or ethyl acetate-based extraction [15]. Seeds of *L. acidissima* with fatty acid and tocopherol content have been recognized as a potent natural antioxidant [14]. Methanolic extract from *L. acidissima* L. fruit pulp has been specifically suggested for efficacious wound healing activity, which was associated with its anti-inflammatory effect [16]. The antioxidant activities of the fruits of *L. acidissima* L. have a high correlation with the contents of flavonoids and phenols [15]. Moreover, *L. acidissima* seeds have been found to contain alkaloids, flavonoids, phenols, terpenoids, tannins, fats, sterols, saponins, glycosides, gums, mucilage, and fixed oils [14]. Another bioactivity, such as antiproliferation activity, has been reported on the methanol-macerated *L. acidissima* L. fruit against human breast cancer cells [17]. In addition, a previous study has reported a broad antibacterial effect of ethanolic extract from *L. acidissima* L. leaves [18]. However, there are no detailed reports on the benefits of *L. acidissima* L. fruit extract in ameliorating hyperuricemia. In the previous literature, the *L. acidissima* L. fruit has been investigated as an antihyperuricemic agent, but the study failed to induce the animal model into the hyperuricemic state [19]. A study reported the potential of *L. acidissima* L. fruit in promoting the dissolution of kidney stones [20]. Owing to its strong antioxidant potential that could be utilized to treat hyperuricemia, this study aimed to analyze the effect of *L. acidissima* fruit extract in reducing serum uric acid level and URAT1 expression in potassium oxonate-induced hyperuricemic rats.

## 2. Results

### 2.1. Phytochemical Profile

The phytochemical profile based on gas chromatography-mass spectrometry (GC-MS) has been presented in Table 1. The phytoconstituents presented therein were identified based on the mass spectra compared to those in the National Institute of Standards and Technology (NIST) library. A phytosterol ɣ-sitosterol was revealed as the most predominant phytoconstituent with a peak area = 27.62%. The second most abundant content was an ergosterol—3β-ergost-5-en-3-ol (peak area = 11.61%). Lanosterol and lupeol, members of the triterpenoid class, were identified in the extract. Fatty acids, 11,14-eicosadienoic acid, and oleic acid were among the phytoconstituents. Stigmasterol and stigmasterone were captured in the chromatogram with peak area percentages of 13.35% and 0.31%, respectively. The GC chromatogram depicting the emergence of peaks belonging to ɣ-sitosterol and 3β-Ergost-5-en-3-ol along with their respective mass spectra have been presented in Figure 1.

### 2.2. Antioxidant Profile

Antioxidant activity (based on 2,2-diphenyl-1-picrylhydrazyl—DPPH scavenging activity) and capacity (TPC and TFC) of ethanolic extract from *L. acidissima* fruits have been presented in Table 2. The antioxidant activity of the extract, as expressed in IC_50_, was higher than ascorbic acid (0.14 ± 0.02 versus 4.19 ± 0.11 mg/L). The TPC and TFC of the fruit extract were 143.9 ± 5.24 mg GAE/g extract and 390.2 ± 3.66 mg QE/g extract, respectively.

### 2.3. Serum Uric Acid

Changes in the levels of serum uric acid before and after the treatment have been presented in Figure 2. In normal rats, the levels of uric acid were low, and they experienced no significant change at the end of the investigation. Following the potassium oxonate injection, the serum uric acid level was raised to 3.540 ± 0.8081 mg/dL and maintained at a similar level at the end of the study (3.520 ± 0.5495 mg/dL). A small statistical significance in serum uric acid reduction was achieved in the allopurinol group. Decreases in serum uric acid levels in the 200 mg and 400 mg groups were statistically significant with *p* = 0.0091 and 0.0004, respectively.

### 2.4. Kidney Parameters

Effects of the treatment against renal kidney function were determined based on changed levels of blood urea nitrogen (BUN) and serum creatinine (Table 3). No significant change was observed in all groups before and after the treatment, except in the 400 mg group. The BUN level was elevated from 17.60 ± 3.29 to 22.80 ± 3.56 mg/dL with a statistical significance of *p* = 0.007. It is worth noting that dropouts were made for the rats, as the severe illness or injuries were not observed. In the case of serum creatinine, no significant change was observed before and after the treatment in the six studied groups (Table 3).

### 2.5. Liver Parameters

Liver parameters, including aspartate aminotransferase (AST) and alanin aminotransferase (ALT), have been observed and presented in Table 4. The elevation of both liver parameters was observed once the potassium oxonate was injected. The administration of allopurinol significantly reduced serum AST from 224.4 ± 47.68 to 151.2 ± 22.07 IU/L (*p* = 0.198). The reduction of ALT also occurred in the allopurinol group, but statistical significance was not reached (*p* = 0.188). Both AST and ALT levels were reduced significantly in the 400 mg group (*p* = 0.01 and 0.003, respectively), while significant reduction in the 200 mg group was only obtained in the ALT parameter (*p* = 0.005). It is worth noting that AST was reduced in the 200 mg group but with a small statistical significance (*p* = 0.068).

### 2.6. Expression Profile of URAT1

The relative expression of urate transporter 1 (URAT1) among untreated and treated groups have been presented (Figure 3). The relative expression was obtained from the normalization based on endogenous β-actin. The attenuation of URAT1 expression of a 1.23 ± 0.12-fold change was observed in rats injected with potassium oxonate as compared with the normal group. The allopurinol group had a 1.51 ± 0.02-fold change, which was higher as compared with the control (hyperuricemic) group. A decreasing trend of URAT1 expression to 1.31 ± 0.06- and 1.02 ± 0.05-fold changes was observed in the groups treated with 100 and 200 mg of the ethanolic extract from *L. acidissima* fruits. The URAT1 expression rebounded after the administration of the 400 mg extract.

## 3. Discussion

Herein, we reported the antihyperuricemic activity of ethanolic extract from *L. acidissima* fruits as indicated by significant reductions of serum uric acid in the 200 and 400 mg groups. In this present study, the extract contained triterpenoids (lanosterol and lupeol), phytosterol (stigmasterol and ɣ-sitosterol), and fatty acids (11,14-eicosadienoic acid, and oleic acid). Previously, lanosterol and its derivatives have been identified in fungi with serum uric acid-reducing activities via xanthine oxidase inhibition [21,22]. Lupeol and stigmasterol were present in the ethanolic extract of *Lychnophora pinaster*, which have been reported to have antihyperuricemic and anti-inflammatory effects [23]. The inhibition of hepatic and serum xanthine oxidases by ethanolic extract of *Cudrania tricuspidata* leaves containing stigmasterol and β-sitosterol (a stereoisomer of ɣ-sitosterol) in vitro has been witnessed previously [24]. In a previously published report, the presence of phytosterol was observed in petroleum ether and diethyl ether extracts but not in ethanol extract [25]. The findings from the previous study were based on a simple qualitative analysis, while this present study used semi-quantitative analysis based on GC-MS. Therefore, the results from this present study are a novel finding, providing a more detailed understanding of the phytoconstituents of the ethanolic extract from *L. acidissima* fruits.

In this present study, the presence of fatty acid content was found in the extract, which possibly assists the urate excretion. Fatty acids were found to improve uric acid levels via xanthine oxidase inhibition and regulation of renal urate reabsorber, URAT1 [26,27]. In a previous study, *L. acidissima* pulp has been reported to contain monounsaturated fatty acids, including oleic acid [14]. Oleic acid could improve nephrotic injury by acting as an anti-inflammatory agent [28]. Taken altogether, fatty acids along with other bioactive constituents in the fruit of *L. acidissima* could contribute to the reduction of serum uric acid levels.

Other than serum uric acid, AST and ALT were found to be ameliorated in the 200 and 400 mg groups. In the allopurinol group, only AST was found to be significantly reduced. Following the hyperuricemia induction, all rats experienced elevated levels of AST and ALT, indicating its adverse effects against liver function. A population-based study in the United States confirmed that hyperuricemic conditions are strongly correlated with liver dysfunction and its related morbidities [29]. The positive correlation of high uric acid level on liver damage has been reported in a study recruiting patients without alcoholic fatty liver disease [30]. The reduced AST and ALT levels in the 200 and 400 mg groups in this present study suggest the hepatoprotective activities of ethanolic extract from *L. acidissima* fruits. Hepatoprotection, at least in animal models, has been showcased in multiple reports following the administration of plant extracts, which is associated with antioxidant and anti-inflammatory activities of the phytoconstituent [31]. As shown in this present study, the antioxidant activity of the extract is even higher than that of ascorbic acid, which was corroborated by the high proportion of TFC and TPC. Previously, the DPPH-scavenging activities of *L. acidissima* fruit extracts (obtained by using water, methanol, ethyl acetate, and chloroform solvents) were found to be high and having significant correlations with TFC and TPC [15]. In that study, the TFC and TPC were found to be lower, with values only reaching the range of 0.75 to 22 mg GAE/g dry extract and 0.55 ± 0.15 to 2.6 ± 0.34 mg QE/g dry extract [15]. In another study, the fruit pulp extract was shown to possess hepatoprotective activities associated with the increase in indigenous antioxidant enzymes [32].

Herein, we observed that the hyperuricemic control experienced an increase in URAT1 expression, which was further elevated when treated with allopurinol. URAT1 facilitates the reabsorption of uric acid from the renal tubules which attenuates plasma uric acid. In previous research, allopurinol was found to upregulate renal URAT1 mRNA expression in potassium oxonate-induced hyperuricemic rats [33]. The study also suggested an indication of renal tissue injury by observing elevated levels of interleukin (IL)-6 and tumor necrosis factor (TNF)-α mRNA due to allopurinol administration [33]. This might be the reason why a significant reduction in AST was found in the allopurinol group in this present study.

Approved URAT1 inhibitor therapies include lesinurad, probenecid, benzbromarone, and sulfinpyrazone, where their administration has serious side effects such as hepatoxicity, gastrointestinal toxicity, and renal toxicity [34]. As of now, uricosuric agents should not be prescribed to patients with renal injury or renal stones [35]. Thus, researchers have investigated natural compounds such as URAT1 antagonists with higher efficacy and lower adverse effects. Several natural URAT1 inhibitors have been identified, such as nobiletin, naringenin, and hesperetin [36]. Dietary flavonoids (i.e., fisetin and quercetin) have been suggested to reduce uric acid reabsorption by limiting URAT1 function and expression [26]. An alcohol soluble (ethanolic extract) of *Urtica hyperborean* was reported to downregulate the expression of URAT1, hence a significant reduction of serum uric acid in hyperuricemic mice [37]. In this present study, we found URAT1 expression returned to normal when treated with 200 mg of *L. acidissima* extract, which is in accordance with the significant decrease in serum uric acid. However, the URAT1 expression increased again when the extract concentration was increased to 400 mg.

The elevation of URAT1 expression could be associated with the toxicity of the extract dosage. The administration of ethanolic extract from *L. acidissima* fruits at a dosage of 400 mg resulted in a significant increase in BUN, indicating the injured renal tissue. In patients with chronic kidney disease, a condition underlying the serum uric acid retention, BUN, and serum creatinine were found to be increased [38]. Therefore, an increase in URAT1 expression along with BUN levels suggests the renal toxicity of the 400 mg extract dosage. Indeed, the uric acid in the 400 mg group reduced more significantly as compared with those in the 200 mg group. This could be due to the fact that the active compounds in the extract work through other mechanisms (i.e., xanthine oxidase inhibition) or target other urate transporters (i.e., organic anion transporters 1 and 2 and ATP binding cassette G2) [4].

One of the limitations of this present study was the absence of supporting phytochemical characterization, such as high-performance liquid chromatography-mass spectrometry (HPLC-MS). It is noteworthy that fruits may contain a complex composition, including but not limited to phenolic compounds, sugars, organic acids, vitamins, saponins, tyramine derivatives, amino acids, and alkaloids. Additional phytochemical identification using HPLC-MS could be helpful in identifying the foregoing components. Moreover, we did not evaluate the effect of the extracts on xanthine oxidase, which is the key enzyme in the purine catabolism into uric acid. Another limitation was that the toxicity profiles of these extracts were only analyzed based on serum markers of the liver and kidney, while histopathological analysis was not performed.

## 4. Materials and Methods

### 4.1. Materials

Ethanol 96%, dimethyl sulfoxide (DMSO), potassium oxonate, carboxyl methyl cellulose (CMC), and 2,2-diphenyl-1-picrylhydrazyl (DPPH) were analytical grade and obtained from Sigma-Aldrich (Selangor, Malaysia). As for ketamine, xylazine, NaCl 0.9%, and allopurinol, they were pharmaceutical grade and purchased from Kalbe Farma (Jakarta, Indonesia). Otherwise, all stated chemicals were used as obtained from the manufacturers without pretreatment. The solvent ethanol was redistilled before it was used.

The plant specimen was collected from Aceh Besar Regency, Aceh Province, Indonesia (5°29′11.3″ N 95°25′40.3″ E), in December 2020. Species determination was performed by a botanist in the Herbarium Laboratory, Biology Department, Universitas Syiah Kuala, Indonesia, on 15 December 2020 (no: B/674/UN11.1.8.4.TA.00.01/2020). The sample was identified as *Limonia acidissima* L. The fruits were collected and separated from its seeds and peels, and the fruit flesh was oven-dried (40 °C). The sample was crushed into fine powder for extraction.

### 4.2. Extraction

The extraction was carried out at the Pharmacology Laboratory, Faculty of Veterinary Medicine, Universitas Syiah Kuala. The dried powder of *L. acidissima* fruit flesh (500 g) was macerated in a sealed container with 2.5 L of 96% ethanol for 72 h at room temperature (25 ± 1 °C), where the cycle was renewed every 24 h (3 cycles in total). The filtrate was collected and treated in a rotary evaporator (40 °C; 30 rpm) until all the solvent evaporated. The extract was sealed in a dark container until further use. Phytoconstituents of the extract was identified using gas chromatography-mass spectrometry (GC-MS QP2020 NX, Shimadzu, Kyoto, Japan), as described by the previous literature [39]. The GC-MS conditions used in this present study were as follows:
Column oven temperature45 °CInjection temperature280 °CPressure51.5 kPaTotal flow14 mL/minColumn flow1 mL/minLinear velocity 36.2 cm/s

The mass spectra for each separated compound were compared with the data from NIST library.

### 4.3. Animal Treatment

Male *Rattus norvegicus* (*n* = 30) were obtained from the Animal Model Laboratory, Biomedical Research Center, Research Hub, Indonesia. The animals had characteristics of being 12–14 weeks and having 200–300 g of body weight (BW). The acclimation was carried out for 7 days in light-dark cycles at 22 ± 2 °C and fed (protein content: 17%) *ad libitum*. All rats fasted for 6 h and were subsequently grouped into 6 groups (normal, control, allopurinol, and three extract groups), in which each group consisted of 6 rats. The normal group was intraperitoneally injected with saline water (NaCl 0.9%) for placebo control, whilst other groups were intraperitoneally injected with potassium oxonate 250 mg/kg BW (suspended in NaCl 0.9%). The blood sample from each priorly anesthetized rat was drawn from the lateral tail vein to determine the baseline levels of serum uric acid, creatinine, aspartate aminotransferase (AST), alanine aminotransferase (ALT), and blood urea nitrogen (BUN). The blood samples were collected after 1 h of allopurinol or the extract administration. In the allopurinol group, the hyperuricemic-induced rats were treated with a dosage of 10 mg/kg BW in a CMC 0.5% suspension (10 mL) once a day using a nasogastric tube for three days. As for the extract groups, each group received 100, 200, and 400 mg/kg BW dosage of ethanolic *L. acidissima* extract in a CMC 0.5% suspension (10 mL) once a day and labeled as group 100, 200, and 400 mg, respectively. The induction and treatment (allopurinol or *L. acidissima* fruit extracts) with 1 h intervals were performed daily for 3 consecutive days. The rats were euthanized using ketamine (100 mg/kg) and xylazine (20 mg/kg) and dissected for their internal organs. The abdominal aortic blood was collected from each of the anesthetized rats after 1 h of the last treatment. The ethical clearance for this research had been granted beforehand by the ethics committee of the Faculty of Veterinary Medicine, Universitas Syiah Kuala (No. 83/KEPH/XII/2020). The humane endpoints set in this research were significant body weight decrease (around 200 g) and exhibiting behavioral signs of pain or distress. The rats were dropped out if they were severely ill or injured during the experiment.

### 4.4. Determination of Serum Markers

All blood sera were centrifuged upon collection at 3000 rpm for 10 min and subsequently stored at −20 °C before use for analysis. Serum uric acid was determined by enzymatic end-point of uricase, where each serum sample was pipetted into a test tube containing a reagent solution. For this purpose, we used commercial kits, namely the AST-GOT kit B74182714 and the ALT-GPT kit B74182713 (Sclavo Diagnostics International, Siena, Italy) for AST and ALT determinations, respectively. The mixture was homogenized for 10 min and read at 500 nm in a UV-vis spectrophotometer. Meanwhile, the ALT and AST levels were determined by a kinetic enzyme assay following the recommendation from the International Federation of Clinical Chemistry (IFCC).

### 4.5. Determination of URAT1 Gene Expression

The gene expressions of URAT1 were determined based on a qualitative polymerase chain reaction (PCR) based on the reported study [36]. Firstly, the isolation of RNA from the rat’s kidney was carried out with Quick-RNATM MiniPrep Plus (Zymo Research). Thereafter, cDNA was synthesized using ReverTra AceTM qPCR RT Master Mix with the gDNA remover (TOYOBO) cDNA synthesis kit. Reverse and forward sequences of DNA primers for each molecules measured using qualitative PCR have been presented in Table 5. The primers were designed based on the previous literature indexed by the National Center for Biotechnology Information (NCBI, https://www.ncbi.nlm.nih.gov/nuccore/) (accessed on 5 September 2022). Predenaturation was carried out at 95 °C for 1 min before continuation with two other cycles: (1) performed at 95 °C for 3 s with 40 repetitions and (2) at 60 °C for 20 s. The cycle threshold (Ct) was taken to semi-quantify the number of mRNA using Applied Biosystem 7500 v.2.0.6 (Thermo Fisher Scientific, Selangor, Malaysia). The gene expression was derived from 2^−[Ct(URAT1)-Ct(β-actin)]^, where the normalization was based on endogenous β-actin.

### 4.6. DPPH Scavenging Assay

The determination of antioxidant activity of the ethanolic *L. acidissima* fruit extract was performed through a free radical DPPH scavenging assay. DPPH powder was firstly dissolved in ethanol (the extract solvent) until the concentration became 40 μg/mL. Together with 5 mL of the extract in various concentrations (1.5–25 mg/L), the DPPH solution (1 mL) was mixed and homogenized using a vortex mixer. The mixture was incubated at 37 °C for a half an hour before being taken for a measurement using a UV-vis spectrophotometer at λ = 517 nm. The median inhibitory concentration (IC_50_) was calculated by a linear regression of extract concentration versus % inhibition curve. Ascorbic acid was employed as the positive control with the same concentration range.

### 4.7. Determination of Total Phenolic and Flavonoid Contents

The extract sample was taken and diluted in ethanol until the concentration reached 0.001 mg/L, where 200 μL of the extract solution was used to make a mixture along with distilled water (15.8 mL), Folin–Ciocâlteu reagent (10%; 1 mL), and NaHCO_3_ 2% (1.5 mL). Following a 2 h incubation, the mixture was measured using a UV-vis spectrophotometer at λ = 763 nm (maximum wavelength of gallic acid). The total phenolic content (TPC) was expressed as a gallic acid equivalent (GAE).

As for the total flavonoid content (TFC), the ethanol-diluted extract of 0.001 mg/L (3 mL) was mixed with AlCl_3_ 2% (0.2 mL), potassium acetate (0.2 mL), and distilled water (5.6 mL). Incubation of the mixture was carried out for 1 h at room temperature. The absorbance of the mixture was measured afterward using a UV-vis spectrophotometer at λ = 435 (maximum wavelength of quercetin). The TFC was expressed as a quercetin equivalent (QE).

### 4.8. Statistical Analysis

The data obtained in this study were processed statistically in GraphPad PRISM version 9 (GraphPad Software, San Diego, CA, USA). The normality test was based on the Shapiro–Wilk test (α = 0.05). Normally distributed data were analyzed for statistical difference based on a paired *t*-test. Otherwise, the statistical difference was observed based on the *p*-value obtained from the Wilcoxon test.

## 5. Conclusions

Ethanolic extract from *L. acidissima* fruit has a serum uric acid-lowering effect with ALT and AST amelioration in potassium oxonate-induced hyperuricemic rats. URAT1 inhibition was found when the extract was administered at a dosage of 200 mg. Administration with a dosage of 400 mg could induce renal toxicity from the extract. The serum uric acid-lowering effect of the extract might be attributed to the phytoconstituents such as triterpenoids (lanosterol and lupeol), phytosterol (stigmasterol and ɣ-sitosterol), and fatty acids (11,14-eicosadienoic acid, oleic acid). In future research, our group will explore the toxicity of *L. acidissima* extract. Further purification of the ethanolic extract to determine the most active antihyperuricemic compound and URAT1 inhibition should be carried out.

## Figures and Tables

**Figure 1 pharmaceuticals-16-00419-f001:**
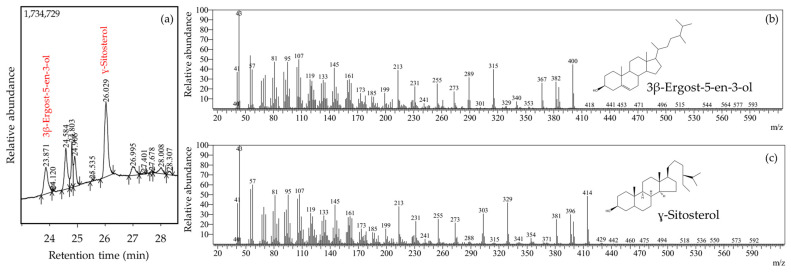
GC chromatogram of ethanolic extract from *L. acidissima* fruits at the retention time range of 23–29 min (**a**). Mass spectra of predominating compounds namely 3β-ergost-5-en-3-ol (**b**) and ɣ-sitosterol (**c**) after their separation from the ethanolic extract from *L. acidissima* fruits via GC.

**Figure 2 pharmaceuticals-16-00419-f002:**
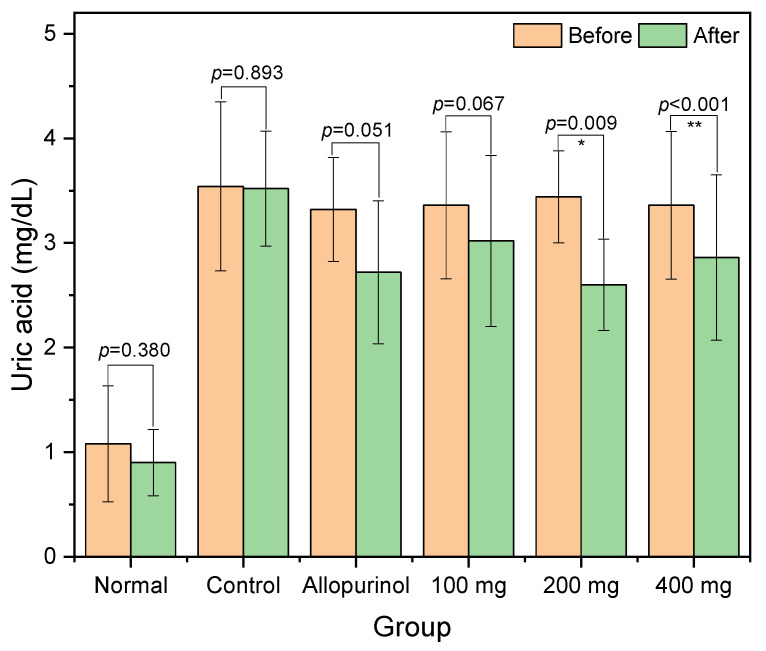
Serum uric acid levels in the hyperuricemic rats before and after the intervention. * Statistically significant at *p* < 0.01 and ** very significant at *p* < 0.01 based on paired *t*-test.

**Figure 3 pharmaceuticals-16-00419-f003:**
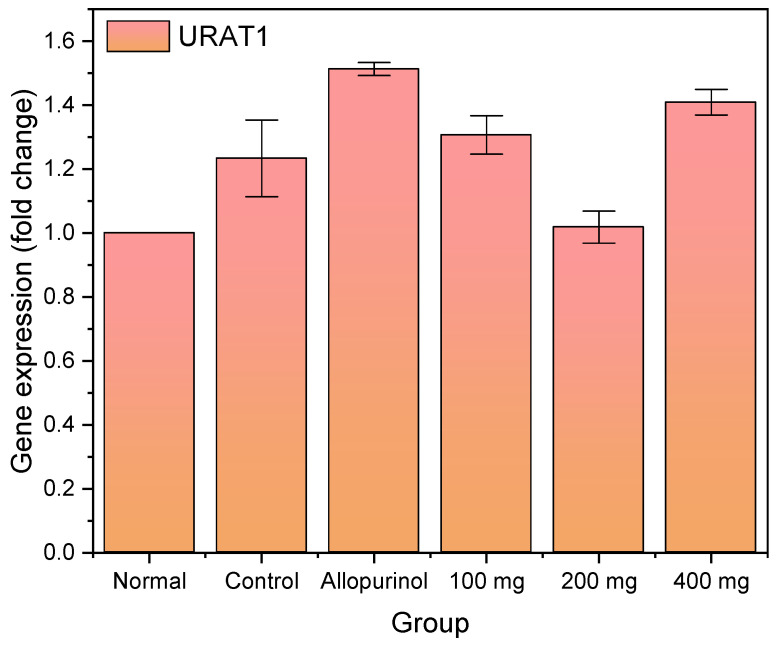
Relative expression of urate transporter 1 normalized with endogenous β-actin.

**Table 1 pharmaceuticals-16-00419-t001:** GC-MS-based phytochemical profile of ethanolic extract from *L. acidissima* fruits.

Peak	Retention Time (Min)	Area (%)	Phytocompound
1	4.025	0.56	(3-Methyl-oxiran-2-yl)-methanol
2	4.090	6.51	2-Pentanone, 4-hydroxy-4-methyl
3	4.305	0.80	Ethylbenzene
4	19.042	0.41	Hexadecanoic acid, methyl ester
5	19.454	3.23	n-Hexadecanoic acid
6	20.799	0.14	11,14-Eicosadienoic acid
7	20.853	0.82	9-Octadecenoic acid (Z)-, methyl ester
8	21.264	4.38	Oleic acid
9	21.481	1.73	Octadecanoic acid
10	23.871	11.61	3β-Ergost-5-en-3-ol
11	24.120	0.22	2′-O-methyl-guanosine
12	24.584	13.35	Stigmasterol
13	24.803	7.93	Bis(2-ethylhexyl) phthalate
14	24.906	7.98	Tris(2,4-di-tert-butylphenyl) phosphate
15	25.535	0.27	N-Decanoylmorpholine
16	26.029	27.62	ɣ-Sitosterol
17	26.995	4.33	Cholest-4-en-3-ol
18	27.401	0.53	N-pentyl-arachidamide
19	27.678	0.31	Stigmasterone
20	28.008	2.64	Lanosterol
21	28.307	1.17	Lupeol
22	29.389	3.46	ɣ-Sitostenone

**Table 2 pharmaceuticals-16-00419-t002:** Antioxidant activity and capacity of ethanolic extract from *L. acidissima* fruits.

Variable	Value, Mean ± SD
TPC (mg GAE/g extract)	143.9 ± 5.24
TFC (mg QE/g extract)	390.2 ± 3.66
IC_50_ of DPPH Inhibition (mg/L)	0.14 ± 0.02

IC_50_ of ascorbic acid = 4.19 ± 0.11 mg/L for DPPH inhibition

**Table 3 pharmaceuticals-16-00419-t003:** Levels of blood urea nitrogen and creatinine before and after the intervention.

Parameters ^a^	Before	After	*p*-Value
Blood urea nitrogen, Mean± SD (mg/dL)			
Normal	23.00 ± 9.35	22.80 ± 8.47	0.704
Control	19.80 ± 5.26	20.60 ± 7.16	0.700
Allopurinol	22.80 ± 10.38	19.40 ± 4.04	0.449
100 mg	19.80 ± 3.77	21.00 ± 2.00	0.468
200 mg	23.80 ± 4.15	21.60 ± 4.83	0.216
400 mg	17.60 ± 3.29	22.80 ± 3.56	0.007 **
Creatinine, Mean± SD (mg/dL)			
Normal ^b^	0.66 ± 0.13	0.68 ± 0.19	>0.999
Control	0.70 ± 0.23	0.62 ± 0.11	0.512
Allopurinol ^b^	0.68 ± 0.19	0.64 ± 0.09	0.750
100 mg	0.54 ± 0.30	0.70 ± 0.16	0.140
200 mg	0.78 ± 0.26	0.74 ± 0.24	0.670
400 mg	0.64 ± 0.21	0.62 ± 0.27	0.621

^a^ Unless otherwise stated, the analysis was carried out using paired *t*-test. ^b^ Analyzed using Wilcoxon test. ** Statistically very significant at *p* < 0.01.

**Table 4 pharmaceuticals-16-00419-t004:** Levels of serum aspartate aminotransferase and alanine aminotransferase before and after the intervention.

Parameters ^a^	Before	After	*p*-Value
Aspartate aminotransferase, Mean ± SD (IU/L)			
Normal	139.2 ± 11.43	140.4 ± 2.07	0.833
Control	208.4 ± 49.13	206.4 ± 55.13	0.742
Allopurinol	224.4 ± 47.68	151.2 ± 22.07	0.020 *
100 mg	197.5 ± 30.40	170.8 ± 43.06	0.107
200 mg	201.4 ± 17.80	162.8 ± 23.26	0.068
400 mg	216.0 ± 21.06	170.2 ± 26.29	0.010 **
Alanine aminotransferase, Mean ± SD (IU/L)			
Normal ^b^	80.0 ± 7.91	80.80 ± 5.85	0.750
Control ^b^	136.2 ± 41.57	133.8 ± 36.95	0.875
Allopurinol ^b^	138.6 ± 82.86	95.60 ± 11.48	0.188
100 mg	145.2 ± 50.54	125.0 ± 28.20	0.352
200 mg	142.0 ± 55.76	122.0 ± 53.44	0.005 **
400 mg	144.8 ± 21.04	94.20 ± 17.77	0.003 **

^a^ Unless otherwise stated, the analysis was carried out using paired *t*-test. ^b^ Analyzed using Wilcoxon test. * Statistically significant at *p* < 0.05 and ** very significant at *p* < 0.01.

**Table 5 pharmaceuticals-16-00419-t005:** Primers used to determine the expression of URAT1 using qualitative PCR.

Molecule	Sequence	PCR Product Size (bp)	NCBI Reference Sequence
β-actin	F: 5′-CCTAAGGCCAACCGTGAAAAGATG-3	219	NM_007393.3
	R: 5′-GTCCCGGCCAGCCAGGTCCAG-3′		
URAT1	F: 5′-TTCATGCCCACCTTCCCCCTC TAC-3′	207	NM_009203.3
	R: 5′-CATCCTCCAGCTGCGCACACCATA-3′		

F: Forward; R: Reverse.

## Data Availability

The data presented in this study are available on request from the first author. The data are not publicly available due to the fact that this study is still ongoing.

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
