# Peer review of "Ethanolic Extract from Limonia acidissima L. Fruit Attenuates Serum Uric Acid Level via URAT1 in Potassium Oxonate-Induced Hyperuricemic Rats"

_pharmaceuticals, 2023, doi:10.3390/ph16030419_

Round 1

Reviewer 1 Report (Previous Reviewer 1)

Authors' objective was to investigate the anti-hyperuricemic activity of the natural product from Limonia acidissima L. in vivo. The extract was obtained through the maceration of L. acidissima fruits using ethanolic solvent and tested for its anti- hyperuricemic activity against potassium oxonate-induced hyperuricemic rats. After the modification, it is better, however, some questions still need to be modified.

1. Please provide standard deviation in the figure 2. These data need to be tested 3 times.

2. GC-MS-based phytochemical profile of ethanolic extract from L. acidissima fruits showd it contains a lot of compounds. Authors shoud disscuss which compounds might exhibit stronger anti-hyperuricemic activity or  play a main role.

Author Response

Reviewer 1

Comment from reviewer: Authors' objective was to investigate the anti-hyperuricemic activity of the natural product from Limonia acidissima L. in vivo. The extract was obtained through the maceration of L. acidissima fruits using ethanolic solvent and tested for its anti- hyperuricemic activity against potassium oxonate-induced hyperuricemic rats. After the modification, it is better, however, some questions still need to be modified.

Response from author: Thank you for your brief summary. We have tried our best to accommodate your concern.

Comment from reviewer: Please provide standard deviation in the figure 2. These data need to be tested 3 times.

Response: The data were obtained in triplicate, the standard deviation has been added. Please refer to the newly revised Fig 2.

Comment from reviewer: GC-MS-based phytochemical profile of ethanolic extract from L. acidissima fruits showd it contains a lot of compounds. Authors shoud disscuss which compounds might exhibit stronger anti-hyperuricemic activity or play a main role.

Response: Please pay attention on the discussion Line 176—177 “triterpenoids (lanosterol and lupeol), phytosterol (stigmasterol and É£-sitosterol), and fatty acids (11,14-eicosadienoic acid, oleic acid).” These are phytocompounds that have been reported for having anti-hyperuricemic activity. Since the reviewer raise a concern about this, we also extend the ‘Conclusions’ (Line 385—388). “Serum uric acid lowering effect of the extract might be attributed to the phytoconstituents such as triterpenoids (lanosterol and lupeol), phytosterol (stigmasterol and É£-sitosterol), and fatty acids (11,14-eicosadienoic acid, oleic acid).”

Reviewer 2 Report (Previous Reviewer 2)

The manuscript was improved by introduction of new data regarding the antioxidant content and activity and by completing the method section. Some of the changes requested during the previous review were done, or were mentioned as limitation of the study.

Author Response

Comment from reviewer: The manuscript was improved by introduction of new data regarding the antioxidant content and activity and by completing the method section. Some of the changes requested during the previous review were done, or were mentioned as limitation of the study.

Response: Thank you for your previous inputs and present positive feedbacks.

Reviewer 3 Report (Previous Reviewer 3)

See my comments in the attached PDF file. 

Author Response

Comments from reviewer: I read the rebuttal of the authors on my review on the first version of the manuscript. I want to state clearly that it has never been my intention to offend the authors. I write this because I have the feeling that not all of my comments were received well by authors. I always try with my reviews to help authors to improve their manuscript. For this reason, my reports are quite detailed and lengthy. This is also emphasized in the very last sentence of my review. I found the first version unacceptable but surely encouraged revision. This has nothing to do with ‘undermining’ the work as the authors state in their rebuttal. In some cases the authors had good reasons to disagree with me and this is fully respected, of course.

All in all, I am glad to see that the authors did use many of the issues that I rose for improvement. The manuscript has become much better, I think. But still, before it can be published, there are a few issues to concern.

Response: Thank you for your kind intention to help improving the manuscript. Your lengthy and detailed comments are never the problems. We are grateful that you have been availing your time checking our manuscript.

Comments from reviewer: The meaning of line 82-84 is unclear. What was the purpose of the synthesized nanoparticles?

Response: We aimed to emphasize the antioxidant potential. But since NPs synthesis is not directly link to the antioxidant potential, we choose of removing the statement. Thank you for your insight on this.

Comments from reviewer: The aim of the study, in line 94-96, can be made more explicit (and as a result clearer). I searched for literature on the possible traditional use of this plant and, as written in my first review, came across the two papers of low quality, by googling. The authors may say more explicitly that reference to traditional use of the plant to treat hyperuremia is absent, and that the plant has been limitedly studied so far. The few reports that exist are of low quality. And then the link can be made between the reported antioxidant activity in relation to hyperuremia, which is now tested for L. acidissima in this study. So, replace the word ‘therefore’ by a phrasing that connects antioxidant activity with anti-hyperuremic effects. The examples of other plants with similar properties and applications, as mentioned, then also fit better in the whole story.

Response: Thank you! This is a good idea to add more sense to the whole story. Supposedly this modified paragraph from the last paragraph of the introduction is better:

“In previous literature, the L. acidissima L. fruit has been investigated as an anti-hyperuricemic agent, but the study failed to induce the animal model into the hyperuricemic state[22]. A study reported the potential of L. acidissima L. fruit in promoting the dissolution of kidney stone [23]. Owing to its strong antioxidant potential that could be utilized to treat hyperuricemia, this study aimed to analyze the effect of L. acidissima fruit extract in reducing serum uric acid level and URAT1 expression in potassium oxonate-induced hyperuricemic rats.”

Comments from reviewer: Table 1 (especially): please reconsider the number of digits behind the decimal point. In my opinion there are too many (see sd!’s), which suggest non-realistic precision. I understand that the authors are reluctant to adapt this and I leave it the editor to decide whether this should be changed or not. In general, for such data in mg/dL (which fine a unit although mmol/L is also used), two digits (or even one for larger values) behind the decimal point would suffice. It is a matter of rounding off. For the statistical analysis no rounding off is done.

Response: We agree about the rounding off. The two digit behind decimal has been presented in the table. For p value it is true that the three digits behind the decimal is the international standard, for instance p<0.001 should be used instead.

Comments from reviewer:  Section 2.5. Please also include how the compounds were identified (based on their mass spectra). Did the authors use reference compounds or a library? Connect with 4.2 (Methods). The word ‘suggested’ is line 343 is too vague. Possible alternative: ‘described’. I would add more information here (principle of the method, identification of compounds).

Response: Agree. It is a good idea to include how the compounds were identified. “The phytoconstituents presented therein were identified based on the mass spectra compared to those in the compound library.” We have also considered to use ‘described’ instead of ‘suggested’

Comments from reviewer: I suggest to adapt the sequence of the results. Start with the phytochemical analysis and antioxidant activity. This is the basis for the further work. This sequence is already used in the Discussion and in the Methods sections.

Response: Thank you for noticing the uniformity of the section sequence. We have amended the manuscript accordingly.

Comments from reviewer: Line 457: the abbreviation QD (quaque die) is found on prescriptions, but it is not appropriate for a methods section in which conditions should be described as clearly as possible. Please replace. Many readers will not be familiar with this ‘pharmaceutical Latin’.

Response: Agreed. We have replaced ‘QD’ with ‘once a day’

In conclusion, I judge the manuscript suitable for publication in Pharmaceuticals, provided that the point mentioned above are considered and addressed.

Response: Thank you.

This manuscript is a resubmission of an earlier submission. The following is a list of the peer review reports and author responses from that submission.

Round 1

Reviewer 1 Report

This study confirmed the anti-hyperuricemic effects of an ethanolic extract from L. acidissima fruit on rats with hyperuricemia brought on by potassium oxonate. This study identified the effective dose and toxic dose of the crude extract using a number of signals, including serum uric acid, creatinine, aspartate aminotransferase (AST), alanine aminotransferase (ALT), and blood urea nitrogen (BUN). The guidance for animal experiments on later purification monomeric chemicals is of minimal value, however, as the experiment employs crude extracts. Please determine its main compontents. 

Comments:

1) Please keep the format consistent: for example, there are spaces before and after the “ = ” in L53, but there are no spaces around the “ = ” in L80.
2) L83 and L155 are missing mass units “levels in 200 mg and 400 groups”.

3) Grammar problem: L57, maybe “with increased level of serum uric acid” should be changed to “with an/the increased level of serum uric acid”.

4) Grammar problem: L60, maybe “one of underutilized medicinal plants” should be changed to “one of the underutilized medicinal plants”.

5) Grammar problem: L65, maybe “have been recognized as potent natural antioxidant” should be changed to “have been recognized as a potent natural antioxidant”.

6)L130:Subheadings should be supplemented with the number ”2.5”.

7) Please add more illustrated materials to Part “Phytochemical profile” of the article: How different retention times relate to different types of compounds.

Reviewer 2 Report

The manuscript submitted by Yusnaini and coworkers investigated the in-vivo (rat) anti-hyperuricemic potential of an ethanolic extract obtained from the fruits of Limonia acidissima.

The most important merit of this manuscript is the originality, as there are no similar studies in the literature dealing with Limonia acidissima extracts. A reduction in the serum level of uric acid and aminotransferases was obtained for 200 and 400 mg extract treated groups, as well as a reduction of URAT1 expression for 200 mg group.

However, there are some important shortcomings which must be addressed before publication.

Here are the major points:

-         The aim of the article is not explicitly stated

-         Lines 56-72 present a series of literature data which are not directly related to the topic of the paper. The authors should present in details literature data regarding the composition and pharmacological effects of fruits, and not of the other parts of the plant.

-         Methods: The methodology is not sufficiently described. Is there a group of animals treated only with the extract? It would be important to see the effect of the extract on hepatic and kidney parameters in normal conditions (without potassium oxonate). Injection – intraperitoneally? The extract was administered every day, for three days? It is not clear

-         Extraction (line 219) is it ethanol:water (1:1)?

-         The extract was not properly characterized. GC-MS is a powerful analytical method but it can be applied only for volatile molecules (or derivatized non-volatile compounds). As the composition of the fruits is a very complex one, including phenolic compounds, sugars, organic acids, vitamins, saponins, tyramine derivatives, amino acids, alkaloids etc. (compounds which are extractable with ethanol), alternative methods such as HPLC-MS should be used for characterization.

-         The chromatographic conditions for GC-MS must be provided.

-         The effect of the extract on the key enzyme, xanthine oxidase, should be assessed

-         Other hepatic markers and histological analysis on liver and kidney should be performed, especially because there were toxic effects at high dosage

-         Tabel 1 and 2 - measurement units are missing

-         Be consistent in using the abbreviation for AST (not SGOT)

Reviewer 3 Report

The study by Yusnaini et al. regarding antihyperuremic activity of an ethanolic extract made from fruits of Limonia acidissima L. is interesting but the manuscript has a number of serious shortcomings.

First, a thorough English language check is absolutely necessary. 

Title

The second part of the title is unclear. Further, I would advise to use a more informative title zooming in on the effect of the extract tested. 

Abstract

The abstract can be more informative by inclusion of the main results (with concrete data).

Introduction

Also mention what the consequences of hyperuricemia are: development of gout, kidney stones.

Results of other natural products, orange juice and the plants mentioned (line 47-55) do not seem so relevant to me. What would be more relevant and is missing, is a description of the background of testing extracts fruit bark of L. acidissima. Is this plant (or the fruits) traditionally used to treat hyperuricemia-related complaints? Now it seems that the authors randomly chose this plant; a rationale is missing. Please add relevant literature here.

In line 60-73 various biological activities with the wood apple are made but they are not linked with (groups of) plant secondary metabolites. Please add. Is there a difference between fruit extracts, aerial extracts? The wound healing effect mentioned in line 68-69, was this fond for aerial parts or fruits? 

Results

The results should start with a thorough analysis of the extracts used. Then the reader knows what it is bout, what the biologically active compounds are. Thus, start with the phytochemical profile (which has no number in the manuscript). The GC-MS analysis only shows the presence of a number of triterpenes and fatty acids. How about phenolic compounds? The have pronounced antioxidant properties. I think the phytochemical analysis is incomplete. This means that the biological effects described cannot unambiguously be attributed to certain classes of secondary metabolites. This makes the story very weak.

In general, when I read about statistics, it is not clear which groups are compared. Do you compare with untreated control, with allopurinol control? Is there a comparson between the doses administered?

In the presentation of the data, mention the group size (n=?). In the methods section it is mentioned that there were dropouts. In Fig. 1 only show significant effects. A p-value >0.05 is just no difference and we are interested in differences.

The data presented in the text and in the figures contain too many digits. Eg., 3.560 mg/dL should be 3.6 mg/dL. Please use proper SI-unit for presenting the data.

For statistics I advise to use p<0.05 and p<0.01 only. There is no added value in giving the exact, calculated p-values. Statistically significant means p<0.05. A ‘thin’ statistical significance sounds odd.

Legend to Fig. 1. Be more concrete about ‘before’ and ‘after’. What is ‘intervention’?

Figure 2: What do you mean with ‘relative expression’? How was normalization done (add to methods).

The results should also include information on dropouts, any observed adverse effects. How well were the extracts tolerated? It is suggested in the methods (line 247-248) that rats became severely ill. This is important information. Should be connected to autopsy.

Discussion

In the first paragraph other plants with antihyperuremic effects are mentioned of constituents of other plant species (triterpenes, fatty acids). Is there no previous work on wood apple that can be added? By simply googling, I was able to retrieve articles from the internet that are not mentioned here. (Shrivastava et al. Plant Arch. 2019, suppl 1; 19: 608 / Veryanti et al. Media Farmasi 2020; 17: 105 (in Bahasa) / Sing et al. IJPSR 2020; 11: 3347.)

What I miss in the discussion is the meaning of the results for the use of wood apple fruits to treat hyperuremia.

Materials and methods

4.1 and 4.2: what exactly was extracted? The combination of seeds, peels and dried fruit flesh?

Use consequently ‘ethanol 96%’ instead of ‘ethanol solvent’.

4.2 How often was the renewal of extraction repeated? Also, give the amounts of plant material used. 

4.2 Phytoconstituents (line 222): be concrete. The GC-MS method applied should be described shortly plus the way the compounds were identified. What do you mean with ‘as suggested by the previous literature’?

4.3 Line 232: mention group size. Apparently, this is 30:6 = 5.

How was the injection administered (intraperitoneally?)?

I assume that the rats were anaesthetized when taking blood samples?

What was the administration scheme of all drugs given? In other words, how much time was there between giving potassium oxonate and the test substances? 

It is not clear to me how and how frequently the extracts were given. What do you mean with ‘using a nasogastric tube for three days’? Did the rats receive three doses, each day one? What is QD?

I don’t understand the meaning of line 245, human endpoints.

Give data about dropouts in the results.

4.4 Add appropriate references and more data. Reagent solution? Did you use a kit? If so, provide details.

4.5 Text says Table 4, Legend says Table 5. How were the primers designed? 

4.6 Line 279: what do you mean with ‘otherwise’? Not-normally distributed data sets?

Conclusion

This is not really a conclusion, but rather a part of the abstract. Here, more concrete data are given. I advise to transfer those to the abstract. What do your results really mean? The authors hint to future work on toxicity, but is I worth to continue into that direction. I would first try to get more insight into the compounds in the extract and their contribution to the effect. If you would, in the future, go for a standardised product (fytofarmaka!), I should be known on which compounds. 

References

Refs 1, 21 and 31 are incomplete (volume and pages missing). DOIs to be added to a number of references. Use capitals in titles and journal names consequently.

Based on the above I do not recommend the manuscript to be published in Pharmaceuticals. Hopefully, my comments and suggestions are useful for preparing a new manuscript.

Reviewer 4 Report

  manuscript entitled "Ethanolic extract from Limonia acidissima L. fruit as anti-hype- 2 pipemic agent: A study using potassium oxonate-induced 3 Wistar rat model ' is on the prevalence of hyperuricemia among adults and the elderly. The aim of the study
was to investigate the hyperuricemic antiactivity of
a natural product from Limonia acidissima L. in vivo. The extract was obtained by maceration of L. acidissima fruit with an ethanolic solvent and tested for
hyperuricemic antiactivity against
potassium oxalate hyperuricemic rats. Presented evidence of L. acidissima fruit extract lowering serum uric acid along with improved AST and ALAT. The decrease in serum uric acid concentration followed a downward trend of URAT1, except for the group treated with 400 mg/kg body weight extract. At the same time BUN increased significantly in the 400 mg group, suggesting renal concentration toxicity. I believe that the manuscript was prepared correctly. An interesting topic was taken up.
The whole manuscript is well composed. The literature used is new and sufficient. However, a few points could be added to improve the manuscript:

line 65:  Are only fatty acids and tocopherols antioxidants in this plant?

line 66: which affects the antioxidant potential?

  I believe that in the 'results' section a table could be provided
with the exact antioxidant compounds and their content in the extracts.